# Impact of Peptide Transport and Memory Function in the Brain

**DOI:** 10.3390/nu16172947

**Published:** 2024-09-02

**Authors:** Lihong Cheng, Caiyue Shi, Xixi Li, Toshiro Matsui

**Affiliations:** 1Department of Cell Biology, Graduate School of Medical and Dental Science, Tokyo Medical and Dental University, Tokyo 113-8519, Japan; cheng.lihong@tmd.ac.jp; 2Department of Bioscience and Biotechnology, Faculty of Agriculture, Graduate School of Kyushu University, Fukuoka 819-0395, Japan; s.caiyue@agr.kyushu-u.ac.jp (C.S.); l.xixi@agr.kyushu-u.ac.jp (X.L.)

**Keywords:** peptide, bioavailability, dementia, Alzheimer’s disease, blood-brain barrier, cognitive impairment

## Abstract

Recent studies have reported the benefits of food-derived peptides for memory dysfunction. Beyond the physiological effects of peptides, their bioavailability to the brain still remains unclear since the blood-brain barrier (BBB) strictly controls the transportation of compounds to the brain. Here, updated transportation studies on BBB transportable peptides are introduced and evaluated using in vitro BBB models, in situ perfusion, and in vivo mouse experiments. Additionally, the mechanisms of action of brain health peptides in relation to the pathogenesis of neurodegenerative diseases, particularly Alzheimer’s disease, are discussed. This discussion follows a summary of bioactive peptides with neuroprotective effects that can improve cognitive decline through various mechanisms, including anti-inflammatory, antioxidative, anti-amyloid β aggregation, and neurotransmitter regulation.

## 1. Introduction

As the global population ages, the prevalence of central nervous system (CNS) disorders such as Alzheimer’s disease (AD), Parkinson’s disease (PD), stroke, anxiety, depression, Huntington’s disease (HD), and amyotrophic lateral sclerosis (ALS) is increasing [1]. These conditions contribute to 12% of worldwide annual fatalities, presenting significant challenges to healthcare infrastructure and causing substantial emotional and financial strain on affected individuals and their families [2]. Compared to therapeutic strategies for drug treatment, the development of preventive foods against CNS disorders is among the most important intervention methods. Bioactive peptides are short fragments of proteins, typically containing 2–20 amino acids, known for their advantageous physiological effects [3] against diseases, such as neuroprotection, memory improvement, and anti-hypertensive, anti-microbial, anti-thrombotic, antioxidant, anti-cancer, and osteoprotective effects [4,5]. Currently, there have been in vivo human and animal studies on improving impaired memory and ameliorating neurodegeneration through daily intake of peptides [6,7,8,9]. However, peptides are often unstable and can be rapidly degraded by the digestive system or bloodstream enzymes before reaching the target tissues, thereby reducing their effectiveness [10]. Therefore, it is important to promote their transportation and absorption in the body, especially in the CNS, because the blood-brain barrier (BBB) system strictly regulates the transport of substances into the brain.

The BBB is a selective permeability barrier that protects the brain from potentially harmful substances in the bloodstream while allowing essential nutrients to penetrate. It comprises tightly packed endothelial cells lining the blood vessels in the brain, along with astrocyte endfeet and pericytes, which together create a highly selective filter [11] (Figure 1). In contrast to astrocytes and pericytes, microglia, which are the innate immune cells of the brain, are also reported to be involved in BBB functional integrity. Chronic microglia activation often leads to sustained inflammation that compromises BBB integrity, which is often associated with neuroinflammatory conditions such as AD [12]. The presence of the BBB makes the development of drugs that target the CNS exceptionally difficult. It is estimated that >98% of small molecules and nearly all large molecules cannot cross the BBB [13,14]. Early reports indicated that only lipophilic molecules with molecular weights of 400–500 Da can cross the BBB via passive diffusion [15]. In 1975, Kastin et al. reported that melanocyte-stimulating hormone (MSH) could penetrate the BBB of mice using radioisotope-labeled techniques, providing the first evidence that peptides could be transported across the BBB [16]. An increasing number of studies have reported that peptides can penetrate the BBB and exert various effects on the CNS [17]. However, the mechanism of peptide transport across the BBB and the role of peptides in the target brain tissue remain unclear.

Here, we discuss the transportation behavior of peptides that cross the BBB and comprehensively review the preventive mechanism(s) of peptides against memory and cognitive impairment.

## 2. Transport of Peptides across the BBB to the Brain

Despite the difficulties associated with substrates crossing the BBB, there are several ways to transport them to the brain. Transportation routes in the BBB involve passive diffusion, carrier-mediated transport, receptor-mediated transcytosis, and adsorption-mediated transcytosis [18] (Figure 2).

### 2.1. Passive Diffusion

Passive diffusion is an energy-independent pathway that is driven by a penetrant concentration gradient between the luminal and abluminal sides. Transportation occurs between epithelial cells (paracellular diffusion) and epithelial cells (transcellular diffusion) [18] (Figure 2). Tight junctions (TJs) are critical structures that control the movement of hydrophilic substances inside and outside paracellular diffusion pathways. Because of the severe restriction of TJs, BBB transport through passive diffusion is negligible [19]. Contrastingly, lipophilic or hydrophobic substrates of <400 Da can diffuse across the BBB because of their affinity for lipid bilayers of the cell membrane via the transcellular diffusion pathway [20]. Hydrophobic peptides, such as diketopiperazines (DKPs) [21], *N*-methyl phenylalanine oligomers [22], and phenylproline tetrapeptides [23], which can cross the BBB via passive diffusion, have recently been considered as a novel family of brain delivery systems (BBB-shuttles) to transport drugs and other cargoes that cannot cross the BBB. For example, Teixidó et al. designed DKP-cargo constructs to transport dopamine and baicalin across the BBB via a non-competitive passive transport mechanism [21]. Thus, the passive diffusion route for the uptake of interest into the brain is an appropriate approach that does not consider the affinity with receptors or transporters.

### 2.2. Carrier-Mediated Transport

The magnitude of carrier-mediated transport is determined by intermolecular interactions between substrates and transporter proteins expressed on the luminal and/or abluminal membranes of brain capillary endothelial cells. Additionally, the recognition of substrates by transporters determines their influx and/or efflux direction [24]. In an energy-dependent or -independent manner, carrier-mediated transport is divided into active and facilitated diffusion transports (Figure 2). In facilitated diffusion, a solute binds to a transporter, triggering a conformational change that allows substances to be carried across the BBB without energy consumption, and the transportation of glucose, galactose, or mannose by glucose transporters 1, 3, or 14 (GLUT1,3,14), nucleotides by equilibrative nucleoside transporter 1 (ENT1), and thyroid hormones via monocarboxylate transporter 8 (MCT8) are mediated [25]. Active transport is an energy-dependent process that moves substrates across a membrane. Active transporters expressed in brain capillary endothelial cells are categorized into two superfamilies: ATP-binding cassettes (ABC) and solute carriers (SLCs) families [26]. Influx-directed transport of substrates is conducted mainly by the SLC family, whereas efflux transport often occurs via the ABC family, except for some SLC transporters such as novel organic cation transporters (OCTN) and excitatory amino acid transporters (EAAT), which are involved in bidirectional and efflux routes [27]. Among the SLC family members, the large neutral amino acid transporter (LAT1, SLC7A5) is the most abundant amino acid carrier and is selectively expressed on the luminal and abluminal membranes of brain capillaries [28]. LAT1 can mediate the transport of drugs such as L-dopa, melphalan, gabapentin, and baclofen to the brain and amino acids such as asparagine, histidine, isoleucine, tryptophan, and tyrosine [29,30]. Cationic amino acid transporters (CATs), including CAT1/3 (SLC7A1/A3), which are primarily expressed on the luminal side of brain endothelial cells, mediate the transport of cationic amino acids, such as arginine, lysine, ornithine, and homoarginine [31,32]. Proton-coupled oligopeptide transporters, including peptide transporters 1/2 (PepT1/2, SLC15A1/A2) and peptide/histidine transporters 1/2 (PHT1/2, SLC15A4/A3), are involved in the transport of di-and tri-peptides or peptidic drugs [33]. For example, carnosine is transported by PepT2 [34] and glycyl-sarcosine (Gly-Sar) through PHT1 [33,35]. Overall, 287 SLC genes have been identified in the brain, particularly in cells that comprise the barriers and parenchymal cells responsible for transporting various substrates [36]. Thus, carrier-mediated transport primarily facilitates the transcellular transport of small-molecule substances across the BBB.

### 2.3. Receptor-Mediated Transcytosis

Receptor-mediated transport involves interactions between substrates (ligands) and receptors in brain microvascular endothelial cells (BMECs). This interaction promotes the formation of endocytic vesicles, which transport ligands across the BBB via exocytosis for release into the CNS [37] (Figure 2). Receptor-mediated transport is a complex process characterized by its energy-dependent bidirectional nature, which involves the clustering of receptor-ligand complexes, endocytosis, transcytosis, and exocytosis of transported molecules [38]. This pathway is the primary route for delivering peptide hormones such as insulin, epidermal growth factor, glucagon, vasopressin, atrial natriuretic polypeptide, and carrier proteins that transport nutritional and regulatory substances such as transferrin, low-density lipoprotein (LDL), transcobalamin, and other regulatory proteins [39]. These transporters are mediated by receptors such as insulin (IR), transferrin (TfR), leptin, LDL (LDLR), and LDLR-related protein 1 (LRP1) receptors. These receptors are highly expressed on the luminal side of endothelial cells and demonstrate efficient and specific endocytosis or transport, thereby maintaining the physiological activity of the transmitter [40]. Exogenous polypeptides, such as aprotinin, apolipoprotein E, lipoprotein lipase, factor XIa, and Angiopep-2, are appropriate ligands for LRP1 [41].

### 2.4. Adsorptive Transcytosis

Adsorptive transcytosis, referred to as adsorption-mediated transcytosis, is a transcellular uptake process, which is a receptor-independent endocytosis process driven by electrostatic interactions between positively charged peptides and negatively charged surfaces of BMECs [42] (Figure 2). Cationic proteins such as avidin [43], histone [44], protamine [45], and wheat germ agglutinin [46] are transported via adsorption-mediated transcytosis. Furthermore, recent research has provided increasing evidence for advancements in adsorption-mediated transcytosis for drug delivery. The TAT peptide (YGRKKRRQRRR-NH) was the first cationic cell-penetrating peptide identified via adsorption-mediated transcytosis [47]. Based on this finding, TAT-conjugated nanoparticle techniques have been applied for enhanced drug delivery of doxorubicin [48] and paclitaxel [49]. SynB with a cationic sequence has been used as a delivery enhancer for benzylpenicillin or morphine-6-glucuronide transport, without considering the degradation of BBB integrity [50,51]. The conjugation technique using transportan 10 (TP10), a 21-residue amphipathic peptide, can enhance vancomycin delivery via adsorption-mediated transcytosis [52].

Adsorption-mediated transcytosis combined with a peptide-conjugation technique with interest is a novel strategy for appropriate “drug” transport to the brain. Instead, carrier-mediated transport and receptor-mediated transcytosis should be targeted for specific “bioactive small peptide” transport to the brain.

## 3. Evaluation of Peptide Transportability into the Brain

Currently, there are several methods for evaluating target peptides across the BBB using in vitro BBB models, in situ brain perfusion experiments, in vivo animal experiments, and visualization experiments using positron emission tomography (PET), magnetic resonance imaging (MRI), and single-photon emission computed tomography (SPECT) techniques [53] (Figure 3).

### 3.1. In Vitro BBB Reconstituted Models for Peptide Transport

In vitro reconstituted models of the BBB in mammals have been employed to evaluate the BBB transportability of target peptides. The design is based on the principle of mimicking the structural and functional characteristics of the BBB. Brain endothelial cells are mounted onto the porous membrane of a transwell insert to generate a BBB membrane barrier system, and astrocytes and pericytes were co-cultured to better mimic the in vivo BBB [54] (Figure 3A). Using a BBB reconstituted model, LYLKPR, a fermented yak milk peptide, showed neuroprotective effect in H_2_O_2_ injured cells [55], and apamin (CNCKAPETALCARRQQH), a venom peptide that shows neuroprotective effects in animals [56], was found to transport this membrane. Cyclo (L-Phe-L-Phe), showing an anti-hypertensive effect, was confirmed to be a penetrant with a high permeability of *P*_app_ of 2.5 × 10^−5^ cm/s through the BBB membrane [57]. A convenient in vitro BBB transport model was explored to investigate the transportability of other oligopeptides, such as PPL [58], αS1-casein peptide (PIGSENSEKTTMPLW) [59], and food-derived hemorphins (H7, LVV-H4, VV-H4, VV-H7) [60], as shown in Table 1. Notably, the model is not an appropriate in vivo BBB system comprising BBB TJ, pericytes, and astrocytes [61], and further studies using brain perfusion experiments, radioactive tracing, and PET measurements are needed to determine the precise BBB transportability of peptides.

### 3.2. In Vivo BBB Transport Models for Peptide Transport

Using a radioactive tracing technique, casomorphin-5 and casomorphin-7 derived from milk were successfully used to transport peptides across the BBB in the brains of mice [62], similar to the significant detection of radiolabeled GTWY [63], LH [64], WY [65], and MKP peptides [66] in animal brains. However, using radioactivity can limit the strength of this conclusion, because the radioactivity detected in each organ can represent the target peptides and their fragmented (metabolized) forms.

In in situ perfusion experiments using animals, the lipoprotein receptor-related protein 1 (LRP1)-binding peptide L57 [41] and ziconotide from sea snails [67] exhibited mouse brain uptake. Significant detection of aprotinin and Angiopep-2 (An2) in the mouse brain was observed after in situ perfusion, which was transported via LRP1 receptor-mediated transcytosis [68] (Table 1). The advantage of in situ perfusion experiments was confirmed by the observed intact transport of soybean dipeptides, GP, and YP (K_i_ value: 3.49 and 3.53 µL/g·min, respectively) in the mouse brain parenchyma, and local accumulation of YP at the hippocampus, hypothalamus, striatum, cerebral cortex, and cerebellum, using an advanced visualization technique by phytic acid-aided matrix-assisted laser desorption ionization (MALDI)–mass spectrometry (MS) imaging analysis [69].

Transport and functional studies of peptides in the brain aim to clarify whether they remain intact after oral intake. Unfortunately, there are only a few reports on the oral administration of dietary peptides, excluding YP [70] and Pro-hydroxyPro [71]. It has been reported that YP orally administered to mice at 10 mg/kg enters the blood circulation with an absorption ratio of 0.15%, following the intact detection of YP in the brain parenchyma with an accumulation ratio of 0.0037% [70]. In vivo improvement of impaired cognitive deficits in working and long-term memory by daily intake of YP in amyloid β (Aβ)-injected acute AD model mice [72] may support the intact absorption of YP from the mouth to the brain. At a high dose of 600 mg/kg collagen hydrolysate in rats, Pro-hydroxyPro was detected in the cerebrospinal fluid, although no pharmacokinetic parameters were available [71] (Table 1).

### 3.3. In Vivo Imaging Techniques for Peptide BBB Transport

In addition to the detection methods mentioned in Section 3.1 and Section 3.2, in vivo imaging techniques such as MRI, PET, and SPECT are also important methods to monitor BBB permeability [73]. MRI is a non-invasive imaging technique that uses powerful magnets, radio waves, and a computer to generate detailed images of the inside of the body, particularly soft tissues such as the brain, spinal cord, muscles, and organs in humans. By employing advanced MRI techniques such as Dynamic Contrast-Enhanced MRI (DCE-MRI), researchers can monitor the transport of contrast agents, which are often conjugated with peptides, across the BBB in real time [74]. André et al. validated the BBB transportation of USPIO-PHO (ultra-small particles of iron oxide (USPIO) functionalized with a disulfide-constrained cyclic heptapeptide (PHO)) via MRI techniques [75]. However, DCE-MRI is generally limited to conditions where the contrast agent easily accumulates in the extracellular space, such as brain tumors, stroke, or multiple sclerosis, and the poor sensitivity and specificity of MRI limits its use in the study of active transport mechanisms [74]. PET and SPECT are molecular imaging techniques combined with specific radiopharmaceuticals that can offer insights into the extent of BBB dysfunction in various neurological disorders. These methods are considered the gold standard for in vivo imaging of transport mechanisms, such as P-glycoprotein (P-gp)-mediated efflux and GLUT1-mediated glucose uptake from the blood [76]. For example, gallium tracers ([^68^Gallium]Diethylenetriamine pentaacetate) have been employed to evaluate paracellular BBB permeability, as these large molecular tracers typically do not cross the BBB under normal physiological conditions, whereas, in cases of epilepsy, insult-induced BBB leakage can be detected using these tracers [77]. In addition, SPECT tracers for brain imaging are categorized into two types: diffusible and non-diffusible. Diffusible tracers, including ^99m^Tc-hexamethyl propylene amine oxime (HMPAO), Xenon-133, and ^99m^Tc-ethyl cysteinate dimer (ECD), could cross the BBB through passive transport and be retained in the brain for enough time, thus permitting image acquisition [78]. Non-diffusible reagents such as ^99m^TcO4-, [^99m^Tc]DTPA, and [^99m^Tc]sestamibi are unable to cross the BBB; therefore, they are used as indicators of BBB integrity [79]. The detection methods of these modalities depend on specific clinical or research needs, balancing factors such as spatial resolution, sensitivity, and the nature of the BBB changes being studied.

## 4. The Effects of Peptides on Alzheimer’s Disease

Neurodegenerative disorders are conditions in which nerve cells in the CNS progressively degenerate and lose their structural and functional integrity, leading to gradual neuronal loss and deterioration of brain and spinal cord function. Neurodegenerative disorders include some of the most significant brain diseases, such as AD, PD, HD, ALS, Friedreich ataxia, Lewy body disease, spinal muscular atrophy, and multiple sclerosis [80]. In this section, we mainly focus on the role of peptides in the pathogenesis of AD.

### 4.1. The Pathogenesis of the Alzheimer’s Disease

There are several hypotheses to explain the pathogenesis of AD, including the amyloid cascade, tubulin-associated unit (Tau) hyperphosphorylation, neurotransmitter imbalances, oxidative stress, and neuroinflammation [81,82]. The deposition of Aβ in the brain parenchyma and cerebral vasculature, along with the presence of intraneuronal neurofibrillary tangles and gradual loss of synapses, are key neuropathological hallmarks of AD. Aβ is generated through sequential proteolytic cleavage of amyloid precursor protein (APP) by two membrane-bound proteases, beta- and gamma-secretases [83]. The generated Aβ peptides tend to aggregate into soluble oligomers that further develop into insoluble fibrils, forming Aβ plaques. An imbalance between continuous generation and clearance efficiency is a crucial factor in abnormal extracellular aggregation [84]. Similar to Aβ plagues, the neurofibrillary tangles (NFTs), which are formed by the hyperphosphorylation of Tau protein, are another important neuropathological hallmark of AD. Tau protein is a microtube-associated protein that is abundantly expressed in neurons of the CNS and cerebral cortex [85]. It was reported that the Aβ accelerates the phosphorylation of the Tau protein, and the toxicity of the Aβ is dependent on the Tau protein [86]. The complexity between the Aβ and Tau protein makes it difficult to develop the AD treatment drugs, and related drug investigations are limited. Recently, two types of anti-amyloid antibody intravenous infusion therapies were approved by the U.S. Food and Drug Administration, including aducanumab and lecanemab, marking the end of nearly two decades without new AD drugs [87]. Instead, commercially available drugs target neurotransmitter systems, such as cholinesterase inhibitors (donepezil, rivastigmine, and galantamine), to increase acetylcholine (ACh) or glutamate receptor antagonist (memantine) levels to reduce the excitotoxicity induced by glutamate in the brain [88].

The occurrence of oxidative stress and neuroinflammation are also well studied to involve in the development of AD. Growing evidence indicates that extensive oxidative stress is a hallmark of AD brains, alongside the well-established presence of senile plaques and NFT [89]. The resulting oxidative stress has been linked to Aβ- and Tau-induced neurotoxicity. Additionally, evidence suggests that oxidative stress may increase the production and aggregation of Aβ and facilitate the phosphorylation and polymerization of Tau, creating a vicious cycle that drives the initiation and progression of AD [90]. By the 1990s, it was widely acknowledged that inflammation was only a result of some neurodegenerative diseases, and the CNS did not easily experience inflammation. In the 1990s, several studies found that long-term administration of anti-inflammatory drugs in individuals reduced AD risk by 50% [91]. Along with research regarding neuroinflammation, it has also been demonstrated to have a strong link with oxidative stress, cellular damage, mitochondrial dysfunction, formation of plagues and NFTs [92].

AD is a complicated disorder involving multiple pathological processes. These processes interact with each other, exacerbating the condition progressively over time, and finally causing cognitive impairment and even death. Unfortunately, the treatment options for AD are limited. Recent approaches in the treatment of AD involve investigating potential molecules from natural products or functional foods with neuroprotective effects and metabolites to modulate signaling pathways associated with the disease. In the present review, we will mainly discuss peptides with neuroprotective activity from functional foods.

### 4.2. Alzheimer’s Disease Prevention by Food Peptides

Plant and animal peptides are known to ameliorate memory impairments related to the AD hypothesis (Table 2). Li et al. reported that papain hydrolysates of soybeans, walnuts, and peanuts exhibited inhibitory activity against H_2_O_2_-induced injury in PC12 cells and improved the recurrent memory ability of normal mice and consolidated memory ability of anisodine-treated mice [93]. Although soybean, walnut, and peanut protein hydrolysates have shown potential as food raw materials for ameliorating neurodegenerative disorders, the specific peptides responsible for their functional properties remain unclear. Further research on the purification and identification of peptides from soy and walnut proteins is required. In 2019, the tetrapeptide VHVV was identified from the flavoenzyme hydrolysate of soybean protein by Ju et al. with neuroprotective potential by upregulating long-term memory-related proteins in spontaneously hypertensive rats [94]. Amakye et al. found that the protein hydrolysates of soybean (PHS), oyster (PHO), and sea cucumber (PHH) had a significant effect on reversing D-galactose-induced aging-related learning and memory impairments and oxidative stress in the following order: PHS, PHO, PHH. Further purification of PHS indicated that WPK and AYLH were active components that strongly alleviated H_2_O_2_-induced oxidative damage in PC12 cells. These results suggest that PHS and purified peptides, WPK and AYLH, have the potential to serve as effective antioxidant agents in functional foods or nutraceuticals aimed at mitigating aging-related learning and memory impairments, as well as oxidative stress [95]. However, the exact mechanism should be further investigated in the future. As mentioned in Section 3.2, orally administered YP in mice can be transported across the BBB [69,70] and attenuate Aβ-induced memory impairment [72]. Further mechanistic studies of YP in NE-4C cells indicated that the dipeptide stimulated ACh production via AdipoR1-induced choline acetyltransferase (ChAT) activation [96], which was consistent with the results obtained in amyloid β-induced AD mice [72]. Currently, most of the commercially used drugs are AChE inhibitors, like donepezil or rivastigmine [88]. The mechanism of the beneficial effects of YP on the brain provideds a new direction for the development of AD-related drugs. The walnut derived peptides (GGW [97], VYY [97], LLPF [97], EVSGPGLSPN [98], PPKNW [99], LPF [100], GVYY [100], APTLW [100], YVLLPSPK [101], TWLPLPR [101], KVPPLLY [101], FY [102], SGFDAE [102], WEKPPVSH [103], WSREEQERE [104], and ADIYTEEAGR [104]) significantly ameliorated cognitive impairments via multiple mechanisms, including alleviating oxidative stress, showing neuroprotective effects against H_2_O_2_-induced neurotoxicity, reducing Aβ plaques, exhibiting anti-inflammatory effects, and ameliorating cholinergic system damage. The neuroprotective effects of peptides derived from walnuts, reported through various mechanisms, are a noteworthy development. This multifaceted action suggests that walnut derived peptides could play a significant role in neuroprotection, potentially offering therapeutic benefits for neurodegenerative diseases and cognitive impairment. Further research is needed to elucidate the specific mechanisms involved and to establish the clinical efficacy of these peptides in neuroprotective applications. WYPGK, derived from pine nuts (*Pinus koraiensis*), improves scopolamine-induced memory dysfunction in mice by enhancing synaptic plasticity via sirtuin 3 activation [105]. These studies suggest that peptides derived from plants like soybeans, walnuts, and peanuts have valuable potential to ameliorate memory impairment through various mechanisms, making them beneficial dietary components for daily consumption.

Bioactive peptides derived from animal sources are an important research area. For example, the dipeptide LN identified from fish protein hydrolysate exhibited strong β-secretase inhibitory activity (IC50 = 8.82 µM) and significantly decreased the production of Aβ in SH-SY5Y cells, highlighting its potential for mitigating AD pathology [106]. FYY and DW from *Benthosema pterotum* protein hydrolysate [107] and sturgeon protein-derived oligopeptides (KIWHHTF, VHYAGTVDY, and HLDDALRGQE) [108] were developed as functional peptides with anti-Aβ aggregation and/or neuron protection activity, suggesting their application as nutraceuticals for age-related neurodegenerative diseases. Sea cucumbers (NDEELNK [109], FETLMPLWGNK [110], HEPFYGNEGALR [110], and KMYPVPLN [110]) have been reported to exhibit neuroprotective effects by improving the cholinergic system, increasing energy metabolism, upregulating the expression of phosphorylated protein kinase A (p-PKA), brain-derived neurotrophic factor (BNDF), and nerve growth factor (NGF) signaling proteins in PC12 cells (for NDEELNK), alleviating oxidative stress in neuroblastoma cells, and improving survival in C. elegans exposed to neurotoxic paraquat (for FETLMPLWGNK, HEPFYGNEGALR, and KMYPVPLN). The hexapeptide QMDDQ from shrimp [111] and oligopeptides (PAYCS and CVGSY) from anchovy protein hydrolysate [112] have been reported to increase ACh content by reducing acetylcholinesterase (AChE) activity in PC12 cells. Furthermore, QMDDQ showed neuroprotective ability via the activation of the anti-apoptosis and PKA/CREB/BNDF signaling pathways [111]. Similarly, FPF isolated from Antarctic krill increased ACh content, CREB, SYN, and PSD-95 expression levels, and suppressed AChE activity in scopolamine-induced AD mice [113].

In addition to plant- and animal-derived peptides, fermented products are natural sources of functional peptides. From fermented cheese (Camembert), Ano et al. identified a peptide KEMPFPKYPVEP that significantly improved memory impairment in mice and increased the content of dopamine and norepinephrine in the frontal cortex [114]. LYLKPR ameliorated oxidative stress-mediated neuronal injury by inhibiting the NLRP3 inflammasome [55]. Recently, bioinformatics research using Molecular Docking (MD), PeptideRanker, BIOPEP, PeptideCutter, and ToxinPred was employed to predict potential bioactive peptides based on the binding efficiency of a target-specific peptide [116]. For example, MD simulations can provide detailed insights into the interactions between peptides and their target molecules at the atomic level, helping to predict their stability, binding affinity, and overall efficacy. However, the bioactive potential of peptides estimated by MD simulations should be confirmed in cell and animal models before applying them to clinical application. For example, Rafique et al. successfully identified three neuroprotective peptides (DFVADHPFLF, HGQNFPIL, and RDFPITWPW) in oat protein hydrolysates using in silico MD simulations and in vitro peptidomics techniques [115]. The neuroprotective activity of these peptides was confirmed in H_2_O_2_-damaged PC12 cells and in a scopolamine-induced zebrafish model [115].

Although numerous food-derived bioactive peptides have been shown to improve memory both in vivo and in vitro, significant limitations remain. Current studies in animal models, predominantly those using rodents and zebrafish with memory impairments and short-term modeling, cannot fully capture the complexities of human AD. The complexity of human AD, including its multifactorial etiology and long-term progression, cannot be adequately modeled by short-term and simplified animal studies. Moreover, the clinical application of bioactive peptides faces several critical challenges that need to be addressed before these compounds can be considered viable therapeutic options for neurodegenerative diseases like AD. One of the primary concerns is the stability of these peptides. Bioactive peptides are often susceptible to degradation by enzymes in the gastrointestinal tract, which can significantly reduce their effectiveness when administered orally. Even if they survive the digestive process, their bioavailability—i.e., the proportion of the peptide that enters the circulation and reaches the target tissue—can be low, further limiting their therapeutic potential.

## 5. Conclusions and Perspectives

With the increase in human life expectancy, the incidence of neurodegenerative diseases among older adults has also increased. Consequently, the prevention and treatment of these diseases are becoming increasingly critical. A promising strategy for addressing cognitive impairment involves using bioactive peptides. Currently, there is increasing interest in discovering neuroprotective peptides or protein hydrolysates and understanding the mechanisms underlying the beneficial effects of these peptides on brain health and function. Here, we summarize the pathogenesis of AD and discuss food-derived peptides with neuroprotective effects and their mechanisms of action (Table 2). Notably, bioactive peptides were identified based on the results obtained from cell and animal models. Clinical trials may further enhance our understanding of neuroprotective peptides.

Additionally, bioactive peptides that exert their physiological effects in vivo must first survive the digestive processes in the gastrointestinal tract and be effectively absorbed to reach their target tissues. Specifically, crossing the BBB is crucial for treating neurodegenerative diseases; however, crossing the BBB restricts the entry of many substances, potentially limiting peptide efficacy (Figure 1). Several methods, including in vitro BBB models, in situ perfusion, and in vivo mouse models, have been used to evaluate peptide transport across the BBB (Figure 3). Each method has advantages and limitations; for instance, in vitro models are convenient but do not accurately replicate in vivo conditions. Using advanced MS techniques to confirm peptide delivery may be the best approach, as YP reaches the brains of mice [70].

In general, it is essential to thoroughly consider the bioavailability and bioactivity of food-derived peptides. For food-derived peptides, it is critical to ascertain whether they remain intact after gastrointestinal digestion and whether they can be effectively transported to target organs to exert their biological effects. If these bioactive peptides maintain their efficacy, they can potentially be used for the prevention of neurodegenerative diseases. However, for future drug development, a thorough understanding of their mechanisms of action is crucial to evaluate their potential side effects and ensure safety. Overall, the key research areas for peptides focus on their stability during digestion, transport to target tissues, especially across the BBB, and understanding their mechanisms of action. Provided the therapeutic potential of peptides in disease management, advancing the development of bioactive peptides is crucial for improving human health.

## Figures and Tables

**Figure 1 nutrients-16-02947-f001:**
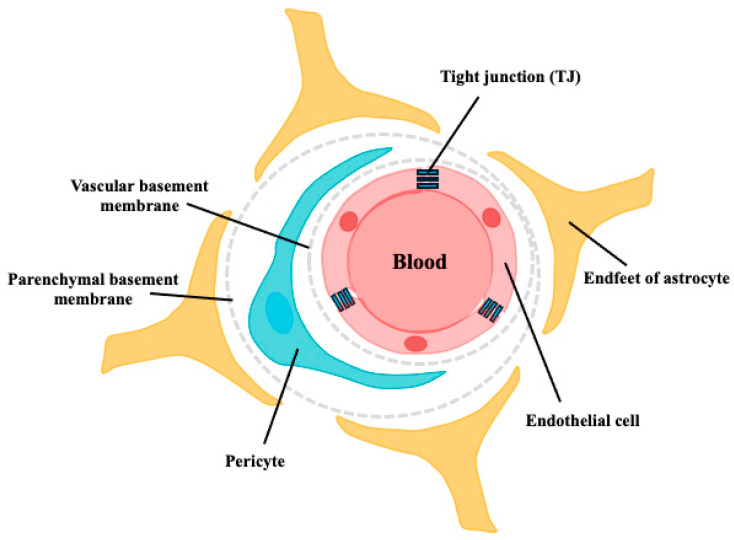
Schematic structure of the BBB. The walls of all brain capillaries consist of a thin monolayer of specialized brain microvascular endothelial cells connected by tight junctions (TJs). These endothelial cells are surrounded by a vascular basement membrane (BM), pericytes, parenchymal BM, and astrocyte endfeet, all of which directly or indirectly contribute to the barrier function of the BBB.

**Figure 2 nutrients-16-02947-f002:**
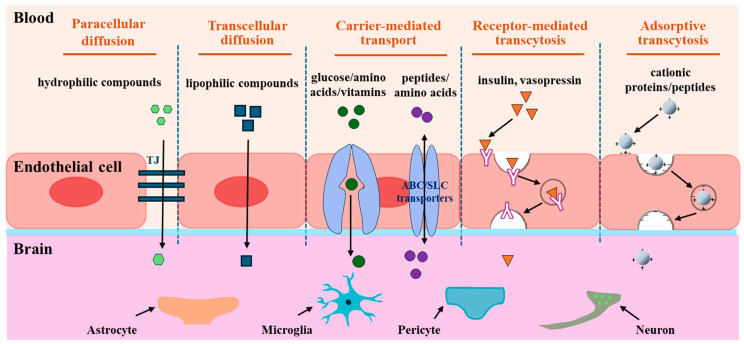
Transport routes across the BBB. The transportation routes include paracellular diffusion, transcellular diffusion, carrier-mediated transport, receptor-mediated transcytosis, and adsorptive transcytosis. Paracellular diffusion is an energy-independent pathway that occurs between epithelial cells, and transcellular diffusion occurs through epithelial cells. Carrier-mediated transport pathway is mediated by the transporter proteins expressed on the luminal and/or abluminal side of the brain capillary endothelial cells. Receptor-mediated transcytosis is an energy-dependent pathway involving the binding of the ligand and the receptor, endocytosis, transcytosis, and exocytosis of transported molecules. Adsorptive transcytosis is a receptor-independent endocytosis process driven by electrostatic interactions between positively charged peptides and the negatively charged surfaces of BMECs.

**Figure 3 nutrients-16-02947-f003:**
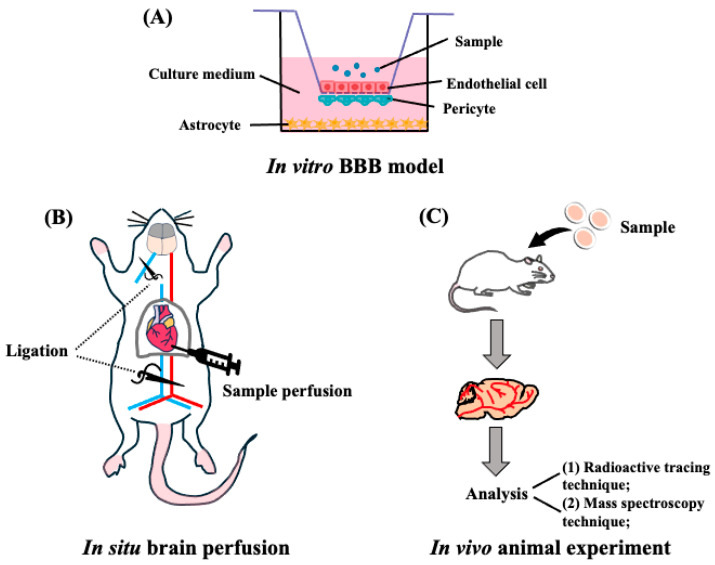
Transportability of peptides across the BBB. (**A**) Schematic representation of **the** in vitro reconstituted BBB model. Endothelial cells are seeded on the upper side of the filter, astrocytes are seeded at the bottom of the plates, and pericytes at the filter membranes of inverted cell culture inserts. (**B**) Representative in situ brain perfusion experiments. After the mice were anesthetized, the descending thoracic aorta was ligated, and at the start of the perfusion, the left jugular was sectioned. After perfusion, the whole brain was removed from the mice and used for detection. (**C**) Schematic representation of in vivo animal experiments. The mouse or rat is administrated with target peptides, then the brain was collected and finally detected by the radioactive tracing technique or mass spectroscopy techniques.

**Table 1 nutrients-16-02947-t001:** BBB transportable peptides reported in the literature.

Peptide	Source	Experiment Model	Transportability	Refs
fermented yak milk peptide		in vitro BBB model	6.90 ± 0.73 × 10^−7^ cm/s	[55]
(LYLKPR)				
apamin	venom	in vitro BBB model	1.7 ± 0.1 × 10^−6^ cm/s	[56]
(CNCKAPETALCARRQQH)				
Cyclo (FF)	chicken essence	in vitro BBB model	~25 × 10^−6^ cm/s	[57]
PPL	fish skin	in vitro BBB model	na	[58]
αS1-casein peptide	milk	in vitro BBB model	1.09 ± 0.14 × 10^−6^ cm /s	[59]
(PIGSENSEKTTMPLW)				
H7	hemoglobin	in vitro BBB model	<1 × 10^−6^ cm /s	[60]
(YPWTQRF)				
LVV-H4	hemoglobin	in vitro BBB model	<1 × 10^−6^ cm /s	[60]
(LVVYPWT)				
VV-H4	hemoglobin	in vitro BBB model	<1 × 10^−6^ cm /s	[60]
(VVYPWT)				
VV-H7	hemoglobin	in vitro BBB model	<1 × 10^−6^ cm /s	[60]
(VVYPWTQRF)				
casomorphin-5	milk	mouse	0.266 nmol/g·min	[62]
(YPFPG)				
casomorphin-7	milk	mouse	na	[62]
(YPFPGPI)				
GTWY	whey protein	rat oral administration (radioactivity)	tissue/plasma ratio: 0.32 (hippocampus) and 0.39 (cerebral cortex)	[63]
LH	-	rat oral administration (radioactivity)	tissue/plasma ratio: 0.2	[64]
WY	fermented dairy products	rat oral administration (radioactivity)	tissue/plasma ratio: 0.23 (hippocampus) and 0.24 (cerebral cortex)	[65]
MKP	milk	rat oral administration	autoradiographic image	[66]
L57	-	in situ mouse brain perfusion	radioactivity	[41]
(TWPKHFDKHTFYSILKLGKH-OH)				
ziconotide	sea snails	rat intravenous injection	0.005%/g brain tissue	[67]
(CKGKGAKCSRLMYDCCTGSCRSGKC)				
Angiopep-2	-	in vitro BBB model and in situ brain perfusion	radioactivity	[68]
(TFFYGGCRGKRNNFKTEEY-COOH)				
Gly-Sar	soybean	in situ mouse brain perfusion	7.60 ± 1.29 μL/g·min	[69]
GP	soybean	in situ mouse brain perfusion	3.49 ± 0.66 μL/g·min	[69]
YP	soybean	in situ mouse brain perfusion	3.53 ± 0.74 μL/g·min	[69]
YP	soybean	mouse oral administration	AUC_0–120 min_: 0.34 ± 0.11 pmol·min/mg-dry brain, at 10 mg/kg	[70]
PO	collagen	rat oral administration	~0.5 nmol/mL in cerebrospinal fluid	[71]
(Pro-Hyp)				

na: not available.

**Table 2 nutrients-16-02947-t002:** Alzheimer’s disease prevention peptides reported in the literature.

Peptide	Source	Experiment Model	Action	Refs
VHVV	soybean	rat	activation of CREB-mediated downstream proteins	[94]
WPK	soybean	PC12 cell	attenuated H_2_O_2_ induced oxidative stress	[95]
AYLH	soybean	PC12 cell		[95]
YP	soybean	mouse and NE-4C cell	stimulation of ChAT expression and ACh production	[96]
GGW	walnut	PC12 cell	protection against glutamate-induced apoptosis	[97]
VYY	walnut	PC12 cell		[97]
LLPF	walnut	PC12 cell		[97]
EVSGPGLSPN	walnut	PC12 cell	protection against H_2_O_2_-induced neurotoxicity	[98]
PPKNW	walnut	APP/PS1 mouse	inhibition of Aβ42 aggregation	[99]
LPF	walnut	mouse	decrease in TNF-α and IL-1β production	[100]
GVYY	walnut	mouse		[100]
APTLW	walnut	mouse		[100]
YVLLPSPK	walnut	PC12 cell	Akt/mTOR-mediated autophagy promotion against oxidative stress	[101]
TWLPLPR	walnut	PC12 cell		[101]
KVPPLLY	walnut	PC12 cell		[101]
FY	walnut	zebrafish	AChE and Keap1 inhibitors	[102]
SGFDAE	walnut	zebrafish		[102]
WEKPPVSH	walnut	BV-2 microglia cell	decreased NO and ROS generation, mitigated secretion of IL-6, TNF-α and IL-1β	[103]
WSREEQERE	walnut	PC12 cell	neuroprotective effect against glutamate-induced apoptosis	[104]
ADIYTEEAGR	walnut	PC12 cell		[104]
WYPGK	pine nuts	PC12 cell and mouse	SIRT3-induced synaptic plasticity enhancement	[105]
LN	pacific hake fish	SH-SY5Y cell	inhibition of Aβ production	[106]
FYY	lantern fish	SH-SY5Y cell and mouse	reduced H_2_O_2_ induced ROS and apoptotic cell death	[107]
DW	lantern fish			[107]
KIWHHTF	sturgeon	RAW264.7	anti-inflammatory effect by inhibiting the MAPK pathway	[108]
VHYAGTVDY	sturgeon			[108]
HLDDALRGQE	sturgeon			[108]
NDEELNK	sea cucumber	PC12 cell	cholinergic system-involved alleviation of cell damage	[109]
FETLMPLWGNK	sea cucumber	human neuroblastoma cell and *Caenorhabditis elegans*	antioxidant activity at both cellular and organism levels	[110]
HEPFYGNEGALR	sea cucumber			[110]
KMYPVPLN	sea cucumber			[110]
QMDDQ	shrimp	PC12 cell	neuroprotective effect by increasing ACh content and inhibiting AChE activity	[111]
PAYCS	anchovy	PC12 cell	AChE inhibition, ROS and Ca^2+^ influx-mediated cell protection	[112]
CVGSY	anchovy			[112]
FPF	Antarctic krill	mouse	elevation of ACh content, AChE inhibition	[113]
KEMPFPKYPVEP	Camembert cheese	mouse	elevation of ACh content, AChE inhibition	[114]
LYLKPR	fermented yak milk	HT-22 cell	amelioration of neuronal injury by inhibiting the NLRP3 inflammasome	[55]
DFVADHPFLF	oat protein hydrolysate	PC12 cell and zebrafish	neuroprotective activity mediated by upregulation of BDNF, Nrf2, and Erg1	[115]
HGQNFPIL				[115]
RDFPITWPW				[115]

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
