# Peer review of "Impact of Peptide Transport and Memory Function in the Brain"

_nutrients, 2024, doi:10.3390/nu16172947_

Round 1
Reviewer 1 Report
Comments and Suggestions for Authors
Comments & Suggestions
In the current article, the authors explore the role of food-peptide transport and its function in the brain, particularly concerning Alzheimer’s disease. While the topic is intriguing, the article requires thorough revision.
A significant limitation of the article is its lack of discussion on the mechanisms of peptide action and the delivery methods.
Major Comments
1. Rewrite the sentence from lines 13-17. Feel free to split it into two separate sentences if needed."
2. In sentences 37-39, clarify the role of the BBB in peptide transportation.
5. In the Figure 2 legend, provide a brief overview of each diffusion method.
6. Combine the two parts of Table 1 into a single table.
7. In table 1, add one column about the role of peptides.
8. In Section 4.2, provide a brief discussion of the mechanisms of action for at least some peptides in the prevention of Alzheimer's Disease, referencing citations 81, 82, and others.
9. In Table 2, include the final results of the studies.
10. Add an additional column to Table 2 for the delivery method or form of the peptide.
Comments on the Quality of English LanguageModerate editing of the English language is required.
Author Response
see attached file for reviewer 3
Reviewer 2 Report
Comments and Suggestions for Authors
The present work stresses the transportation behavior of food-derived peptides with neuroprotective effects that cross the blood-brain barrier (BBB) and highlights preventive mechanisms against memory and cognitive impairments. The work is interesting, however following points need to be addressed to strengthen the manuscript:
The title is general and should be revised focusing on the specific topic.
Regarding the in vitro BBB reconstituted models for peptide transport, please explain the principle they were based and designed. A general comment is that it is not clear enough how peptides focus on their stability during digestion and permeate across BBB and transport to target tissues.
Protease liability plays crucial role in the development of therapeutic peptides and several strategies which have been developed to overcome the limitation of protease-resistant peptides, could be discussed among them non-natural amino acids, chemical modifications and cyclisation. Moreover, the retro-enantio or retro-inverso approach.
How can transportability be measured?
Can the authors discuss the role and selection of specific strategies for effective drug delivery into the brain such as invasive techniques and non-invasive techniques?
Macrophages and microglia interact with the BBB and act against pathogen invasion playing a major role in the innate immune response. The authors could stress the impact of microglia activation on BBB in brain diseases.
Author Response
The present work stresses the transportation behavior of food-derived peptides with neuroprotective effects that cross the blood-brain barrier (BBB) and highlights preventive mechanisms against memory and cognitive impairments. The work is interesting, however following points need to be addressed to strengthen the manuscript:
Comment 1. The title is general and should be revised focusing on the specific topic.
Answer: Thank you so much for your kind suggestions. We revised the title to “Impact of peptide transport and memory function in the brain”.
Comment 2. Regarding the in vitro BBB reconstituted models for peptide transport, please explain the principle they were based and designed. A general comment is that it is not clear enough how peptides focus on their stability during digestion and permeate across BBB and transport to target tissues.
Answer: Thank you so much for your valuable comments. In vitro BBB reconstituted models for peptide transport are designed based on the principle of mimicking the structural and functional characteristics of the BBB. These models typically use endothelial cells that closely replicate the tight junctions and selective permeability of the BBB in vivo. The key aim is to study how peptides interact with and cross this barrier, focusing on their ability to permeate through the endothelial layer, resist enzymatic degradation, and reach target tissues in the brain. We added this part into the manuscript in lines 192-196 as follows.
In vitro reconstituted models of the BBB in mammals have been employed to evaluate the BBB transportability of target peptides. It was designed based on the principle of mimicking the structural and functional characteristics of the BBB. Brain endothelial cells are mounted onto the porous membrane of a transwell insert to generate a BBB membrane barrier system, and astrocytes and pericytes were co-cultured to better mimic the in vivo BBB [53] (Figure 3A).
Comment 3. Protease liability plays crucial role in the development of therapeutic peptides and several strategies which have been developed to overcome the limitation of protease-resistant peptides, could be discussed among them non-natural amino acids, chemical modifications and cyclisation. Moreover, the retro-enantio or retro-inverso approach.
Answer: Thank you so much for your professional suggestions. As you mentioned, protease liability is indeed a critical factor in the development of therapeutic peptides, as it affects their stability and efficacy. Several strategies have been developed to enhance the resistance of peptides to proteolytic degradation including on-natural amino acids, chemical modifications, cyclisation and retro-enantio and retro-inverso approaches. However, in this review, we mainly focus on the BBB-transportable peptides and action mechanisms of brain health peptides about the pathogenesis of neurodegenerative diseases, particularly Alzheimer’s disease. Therefore, we didn’t discuss about the protease liability of peptides in the present review.
Comment 4. How can transportability be measured?
Answer: Thank you so much for your kind suggestions. Several methods are commonly used to evaluate the BBB transportability, such as in vitro BBB models, in situ brain perfusion experiments, in vivo animal experiments, and visualization experiments using positron emission tomography (PET), magnetic resonance imaging (MRI), single-photon emission computed tomography (SPECT) techniques, including TC-99m and metal chelating complexes (Figure 3). These parts were reviewed in chapter 3 “Evaluation of peptide transportability into the brain” in pages 5-7.
Comment 5. Can the authors discuss the role and selection of specific strategies for effective drug delivery into the brain such as invasive techniques and non-invasive techniques?
Answer: Thank you so much for your professional comments. Effective drug delivery to the brain is a complex challenge due to the protective nature of the BBB. Both invasive and non-invasive techniques have been developed to enhance drug delivery to the brain, each with its own advantages and limitations. For example, invasive techniques such as intracerebral injection was used to directly delivers drugs into specific brain regions, bypassing the BBB. It's often employed when other methods are insufficient due to the need for high drug concentrations in specific areas. Non-invasive techniques, for example, nanoparticle-based delivery method, utilizes nanoparticles to encapsulate drugs, enhancing their ability to cross the BBB. It’s Suitable for drugs that need prolonged release or improved stability, and nanoparticles can be engineered to target specific brain cells or regions. The techniques for the drug delivery were not detailed discussed in current review since the content of the present review is to discuss the peptides which can transport across the BBB and the mechanism of action study of the functional peptides regarding to AD.
Comment 6. Macrophages and microglia interact with the BBB and act against pathogen invasion playing a major role in the innate immune response. The authors could stress the impact of microglia activation on BBB in brain diseases.
Answer: Thank you so much for your valuable suggestions. We added the discussion of the microglia activation on BBB in lines 45-48 as follows.
In contrast to astrocytes and pericytes, microglia, which is the innate immune cells of the brain, is also reported to involve in the BBB functional integrity. Chronic microglia activation often leads to sustained inflammation that compromises BBB integrity, which is often associated with neuroinflammatory conditions such as AD [12].
Reviewer 3 Report
Comments and Suggestions for Authors
please see the document

Author Response
I believe that this is an interesting review on two very important issues, the transfer of compounds, and especially peptides, in the brain and their effect on it, especially the protective. Some issues that need to be addressed and to be taken into account are listed below.
Comment 1. I believe the authors should also analyze the peptides that may be of use at other neurodegenerative disorders such as multiple sclerosis, Parkinson’s disease, stroke, Huntington’s disease (HD), and amyotrophic lateral sclerosis (ALS) (in section 4) or change the title and the analysis of sections 4.1 and 4.2 adding information for brain health including other diseases, in order the text to compromise with title of the article.
Answer: Thank you so much for your kind suggestions. In current review, we mainly focused on the functional peptides against memory and cognitive impairments especially for Alzheimer’s disease. Therefore, we discussed about the pathogenesis of Alzheimer’s disease and reviewed the peptides which showed the prevention potential for the cognitive impairments. To avoid the misunderstanding, we revised the title to “Impact of peptide transport and memory function in the brain”.
Comment 2. I believe that since the area of protective peptides for the brain is relatively new. The analysis of the peptides for the most common neurodegenerations could be a very important addition to the manuscript, improving the readability and the citability of the article. Additionally, a more thorough analysis of the protective effect, these added in the new chapters peptides, could have in the brain should be added.
Answer: Thank you so much for your professional suggestions. As you mentioned, neuroprotective effect of the peptides is also an important mechanism in the prevention of Alzheimer’s disease. Therefore, we also included the protective effects of the peptides in section 4.2 and Table 2 such as EVSGPGLSPN (Liu et al., 2019) and WSREEQERE (Wang et al., 2022) from walnut, FYY from lantern fish (Chai et al., 2016), QMDDQ from shrimp (Wu et al., 2020), HGQNFPIL and RDFPITWPW from oat protein hydrolysate (Rafique et al., 2023) and so on.
Comment 3. This new chapters will amplify the extent of the discussion of the text and will provide more and up to date references.
Answer: Thank you so much for your kind comments. Please refer to the answers for comment 2.
Comment 4. In figure 2, the authors bind at the final stage the compounds with astrocytes, I humbly believe this is not always the case. They could generally use the brain structure as a final target.
Answer: Thank you so much for your valuable suggestions. We revised the Figure 2 in page 3 as follows.
Figure 2. Transport routes across the BBB.
Comment 5. “Active transport is an energy-dependent process that moves substrates along a concentration gradient” I believe that it does not necessarily need concentration gradient when it is energy dependent.
Answer: Thank you so much for your carefully reviewing and valuable suggestions. We revised the sentence as follows in lines 112-113.
Active transport is an energy-dependent process that moves substrates across the membrane.
Comment 6. As far as the evaluation transport methods are concerned, I would suggest a more thorough analysis of the techniques first, before the examples of the peptides that have been tested. I also propose the authors to search more information about PET and SPECT techniques, including TC-99m and metal chelating complexes that may have been tested for BBB transport evaluation.
Answer: Thank you so much for your professional suggestions. In this review, we discussed about the properties of the BBB, the BBB transportability of the peptides and the peptides which showed the brain beneficial effects. The BBB transportability of the peptides could be investigated via several techniques. However, we mainly discussed about the results evaluated by the animal levels. As you suggested, we revised the description in lines 172-178.
Currently, there are several methods for evaluating target peptides across the BBB using in vitro BBB models, in situ brain perfusion experiments, in vivo animal experiments, and visualization experiments using positron emission tomography (PET), magnetic resonance imaging (MRI), single-photon emission computed tomography (SPECT) techniques, including TC-99m and metal chelating complexes [52] (Figure 3). In this review, we will mainly discuss about the techniques related to in vitro BBB models, in situ brain perfusion experiments and in vivo animal experiments.
Comment 7. “Brain health of peptides” Do the authors maybe mean “The effects of peptides on brain health”?
Answer: Thank you so much for your kind comments. We revised the subtitle of chapter 4 in line 239 as follows.
- The effects of peptides on brain health
Round 2
Reviewer 1 Report
Comments and Suggestions for Authors
The authors have addressed all the concerns I raised in my previous comments. These modifications have improved the overall quality of the work.
Author Response
Thank you for your reviewing.

Reviewer 2 Report
Comments and Suggestions for Authors
The authors have addressed my comments and suggestions.
Author Response
Thank you for your reviewing.
Reviewer 3 Report
Comments and Suggestions for Authors
The aurhors state "In this review, we will mainly discuss about the techniques related to in vitro BBB models, in situ brain perfusion experiments and in vivo animal experiments."
I believe that the previously suggested techniques are in the area of these experiments, since the in vitro, in situ and in vivo methods concern all the areas of methods. Thus, i believe that more techniques should be added.
Furthermore, the authors use "Table 2. Brain-health peptides reported in literatures", although they want to refer only to Alzheimer and memory impairments. What is the reason for the usage of such a general table since they do not to follow the analysis of more neurodegenerative dieases in their text?
At last, i believe that the extension of the text is two low (i think <5000 words). Thus, i humbly do not think it is able to cover all these aspects that are analyzed, unless it is enlarged.
